# Influence of Structural Characterization of C_3_S-C_3_A Paste under Sulfate Attack

**DOI:** 10.3390/ma16010077

**Published:** 2022-12-21

**Authors:** Qicai Zhao, Tao He, Gaozhan Zhang, Yang Li, Guocheng Rong, Qingjun Ding

**Affiliations:** 1Poly ChangDa Engineering Co., Ltd., Guangzhou 511431, China; 2Advanced Building Materials Key Laboratory of Anhui Province, School of Material Science and Chemical Engineering, Anhui Jianzhu University, Hefei 230022, China; 3State Key Laboratory of Silicate Materials for Architectures, Wuhan University of Technology, Wuhan 430070, China

**Keywords:** C-(A)-S-H gel microstructure, sulfate attack, pore characterization

## Abstract

The durability of C_3_S-C_3_A paste with varied C_3_A content (0%, 5%, 10%, and 20%) against sulfate attack at various attack ages (3 d, 7 d, 28 d, and 180 d) was investigated in this study through the examinations of corrosion product composition, Ca/Si and Al/Si of calcium-(aluminum)-silicate-hydrate (C-(A)-S-H) gel, formation and evolution of microstructure, migration and transformation of Al containing phase products, and pore structure. The results indicated that sulfate attack can promote the hydration reaction in C_3_S-C_3_A paste, thus accelerating the production of C-(A)-S-H gel in the paste. With the increase of C_3_A content, the acceleration effect becomes more significant. In addition, sulfate attack led to the dealumination and decalcification of C-(A)-S-H gel, resulting in the reduction of the gelling power of C-(A)-S-H gel. The degree of dealumination and decalcification of C-(A)-S-H gel increases with the increase of C3A content. At the same time, free Al and Ca promote the formation of expansive products such as ettringite and gypsum. Finally, under the action of sulfate, the pore characterization of C_3_S-C_3_A paste deteriorated, showing a decrease in specific surface area, cumulative pore volume and average pore diameter.

## 1. Introduction

With the development of offshore engineering, more and more building structures have to consider the impact of erosion media. As a widely existing erosion medium in practical engineering applications, the research on the damage of sulfate ions to concrete structures has received extensive attention. Sulfate attack is one of the long-term durability issues in cement-based materials. During the service of concrete structures, sulfate enters into the concrete through diffusion and reacts with cement hydration products Ca(OH)_2_ and C-(A)-S-H to generate the expansive products gypsum and ettringite (AFt), which causes expansion stress in the concrete and accelerates the destruction of concrete. This process reduces the alkalinity of pore solution in the concrete, leading to the instability of C-(A)-S-H gel after dissolution, thus losing the gelling ability [1,2,3,4,5,6]. The mechanical effects on the structural elements can be very serious and can compromise the reliability of the entire construction. As the use of concrete structures is unavoidable in geological sites rich in sulfate, researchers have carried out a series of studies on the performance control of concrete under sulfate attack. C-(A)-S-H, as an important component and main strength source of paste, has received considerable attention in research. Gollop et al. [7] studied the influence of sulfate ion on the microstructure and composition of cement hydration paste, and found that under sulfate attack, Ca^2+^ in C-(A)-S-H would dissolve into the pore solution, promoting the crystallization and development of AFt. Kunther et al. [8] studied the influence of sulfate ion on the microstructure of C-(A)-S-H gel with different synthesized Ca/Si. The results showed that under sulfate attack, interlayer Ca^2+^ in C-(A)-S-H gel would be separated and react with SO_4_^2−^ to form gypsum crystal. The degree of decalcification of C-(A)-S-H gel increases with the increase of the initial Ca/Si. Since the C-(A)-S-H structure is mainly affected by silicon and aluminum phases in clinker, the study of C_3_S-C_3_A paste and its C-(A)-S-H microstructure formed by the cohydration of C_3_S and C_3_A can effectively characterize the hydration process of actual cementitious materials in cement and the formation and evolution mechanism of cementitious-paste microstructure [9,10]. Nicoleau et al. [11] found that SO_4_^2−^ can change the charge on the surface of C_3_S particles by adsorbing on the surface of C_3_S particles, and finally inhibit the charge of C_3_S particles. Al^3+^ can covalently combine with the silica monomer on the surface of C_3_S particles and inhibit the hydration process of C_3_S in a weak alkaline environment.

At present, the formation and evolution of cementitious paste hydration products and their microstructures are mainly studied with cement and other mineral admixtures. Because the hydration processes of multiple minerals in the cement system are coupled and interfere with each other, it is difficult to characterize the hydration process of a single mineral and its influence mechanism on the paste microstructure. In addition, most of the research on the composition and microstructure of C-(A)-S-H gel is carried out by hydrothermal synthesis of soluble Ca salt, silicate, and aluminate, but this method does not involve the hydration process of cement clinker. It is unclear whether its composition and microstructure are the same as that of C-(A)-S-H generated by hydration of actual cementitious materials. Therefore, it is necessary to study the influence mechanism of the type and content of the single mineral of cement clinker on the formation and evolution of the hydrated products and microstructure of the composite hydrated paste, taking the single mineral of cement clinker prepared by the solid-state sintering method as the object.

In summary, C_3_S-C_3_A paste was prepared from C_3_S and C_3_A single mineral synthesized by the solid-state sintering method, and the composite paste with C_3_A substitution rates of 0%, 5%, 10%, and 20% was immersed in 5 wt% Na_2_SO_4_ solution. The composition of erosion products, the formation and evolution of Ca/Si, Al/Si and microstructure of C-(A)-S-H gel, the migration and transformation of Al containing phase products, and the change of pore structure of C_3_S-C_3_A paste at different erosion ages were studied by means of modern testing techniques such as XRD, SEM, TG-DSC, nuclear magnetic resonance (NMR), and hydration heat analysis.

## 2. Experimental Program

### 2.1. Materials

Reagents: calcium carbonate (CaCO_3_ ≥ 99.0 w/%), silicon dioxide (SiO_2_ ≥ 99.0 w/%), aluminum oxide (Al_2_O_3_ ≥ 99.0 w/%), gypsum dihydrate (CaSO_4_·2H_2_O ≥ 99.0 w/%), anhydrous ethanol, and ethylene glycol were provided by Sinopharm Chemical Reagent Co., Ltd. (Shanghai, China).

Water: conductivity was obtained using an ultrapure water mechanism with ≤0.01 μs/cm of ultrapure water.

C_3_S and C_3_A can be efficiently prepared by mixing the above reagents with water using an appropriate quantity.

### 2.2. Preparation of Single Mineral of Cement Clinker

At present, single mineral preparation methods commonly used include the sol gel method, the two-step precipitation method, and the solid-state sintering method, among others. In this experiment, the solid-state sintering method was used. Although this method requires repeated high-temperature sintering, its preparation steps are simple, and the reactants are cheap and easy to obtain which best met the requirements for the preparation of the large number of single minerals in this paper [12].

#### 2.2.1. Preparation of C_3_S

(a)Weigh CaCO_3_ and SiO_2_ at a molar ratio of 3:1. Then put them into a corundum ball mill for mixing for 4 h, during which the rotating speed is kept above 100 r/min.(b)Add a small amount of deionized water to the mixed raw meal to increase its cohesiveness, which is conducive to the next step of pressing and molding.(c)Weigh about 8 g of raw material each time and put it into the mold, then use the press to carry out 50 kN constant pressure and maintain the pressure for 15 s.(d)Calcine the raw meal sheet in a high temperature furnace. The platinum crucible used for calcination can effectively prevent Al in the corundum crucible from mixing into the sintered C_3_S single mineral. Raise the temperature to 1450 °C at a heating rate of 450 °C per hour, and then maintain this temperature for 3 h. Finally, take out the calcined sample and place it in the air for quenching (as shown in Figure 1).(e)Grind the sintered sample and pass it through a 200-mesh sieve.(f)Calcine each batch of C_3_S single mineral sample three times by using the same firing system and use the C_3_S sample powder obtained to prepare a C_3_S-C_3_A paste.

#### 2.2.2. Preparation of C_3_A

The other steps are the same as that of C_3_S single mineral preparation, except that SiO_2_ in step a of C_3_S single mineral preparation is replaced by Al_2_O_3_.

### 2.3. Preparation and Curing of C_3_A-C_3_S Paste

The proportion of composite paste is shown in Table 1. Mix the proportioned CaSO_4_·2H_2_O, C_3_S, C_3_A and deionized water, and pour them into a sealed plastic bottle for molding. The demolding should be carried out after 24 h curing under standard curing conditions (20 ± 1 °C). The demolded samples should be put into sealed plastic bottles filled with saturated Ca(OH)_2_ solution for constant temperature storage. Break the aged test sample into a granular sample with a particle size of about 4–5 mm, and then soak in absolute ethanol solution for 24 h. After the water in the sample is replaced by absolute ethanol, dry it in an oven at 50 °C.

### 2.4. Experimental Program

#### 2.4.1. Determination of Free Calcium Oxide in Single Mineral of Cement Clinker

In this experiment, the ethylene glycol method was used to calibrate the f-CaO content in the synthetic single mineral [13]. The specific operation steps are as follows: put a specified amount of single mineral into the ethylene glycol anhydrous ethanol mixed solution, add the methyl red potassium bromophenol green indicator and stir evenly. Stir at 65–70 °C for 30 min to make f-CaO in single mineral react with ethylene glycol to generate calcium glycol (see Formula 1 for reaction equation). Carry out vacuum suction filtration of the suspended solution and wash the filter residue with anhydrous ethanol three times. Then titrate the filtered filtrate with hydrochloric acid standard solution. When the solution changes from blue to orange, the titration end point is reached. Record the consumption of hydrochloric acid standard solution of the sample.
(1)$$f-\mathrm{CaO}+{\mathrm{CH}}_{2}\mathrm{OH}-{\mathrm{CH}}_{2}\mathrm{OH}={\mathrm{CH}}_{2}O-{\mathrm{CH}}_{2}O-\mathrm{Ca}+H_{2}O$$

Put a specified amount of CaCO_3_ into a crucible and burn it at 950–1000 °C until the mass is constant to obtain CaO for the titration of hydrochloric acid. Take a specified amount of CaO and put it into the ethylene glycol absolute ethanol mixed solution, titrate it in the same way as the sample, and record the consumption of hydrochloric acid standard solution of the calibration sample. Finally, the f-CaO content in single mineral is calculated according to Formulas (2) and (3).
(2)$$T=\frac{m_{\mathrm{CaO}}\times 1000}{V_{\mathrm{CaO}}}$$
(3)$$F=\frac{{T\times V}_{c}}{m_{c}\times 1000}\times 100\%$$
where, T is the CaO titer of hydrochloric acid standard solution, representing the CaO mass corresponding to the unit volume of hydrochloric acid standard solution, mg/mL; m_CaO_ refers to the mass of CaO used for the titration of hydrochloric acid, g; V_CaO_ represents the volume of hydrochloric acid standard solution consumed during titration calibration, mL; m_c_ is the mass of cement single mineral sample, g; V_c_ is the volume of hydrochloric acid standard solution consumed by cement single mineral sample, mL; F is the mass fraction of f-CaO in the single mineral sample, %.

The results of determination of f-CaO content in synthetic single mineral by ethylene glycol method are shown in Table 2. The results show that the f-CaO content of C_3_S and C_3_A single mineral was less than 1% after sintering three times, which can be used to study the formation and evolution mechanism of C_3_S-C_3_A paste microstructure.

#### 2.4.2. X-ray Diffraction Test

D8 ADVANCE X-ray diffraction analyzer produced by Bruker in Germany was used to test the reaction degree of single mineral in the sample and the composition of hydration products. Wherein, the target was Cu(Kα), the rated power was 12 kW, the working current was 100 mA, and the scanning range was (θ) 5°–70° in steps of 0.02°. The XRD diffraction pattern of LUHPC was fully fitted by Jade 6.5 software, and the fitting factor R was ≤7%. Then the phase composition of LUHPC was analyzed using a semiquantitative method (RIR). The test sample was the sample powder passing through a 200-mesh sieve.

#### 2.4.3. SEM–EDS Test

The micromorphology of C_3_S-C_3_A paste was observed with QUANTA FEG 450 field emission environmental scanning electron microscope produced by Hitachi, Tokyo, Japan. The element content of C-(A)-S-H gel area was analyzed to study the Ca/Si change of C-(A)-S-H gel of C_3_S-C_3_A paste. The working voltage was 15–20 kV, the vacuum degree could reach 8 × above 10^−3^ Pa, and the magnification was 1000–5000 times. The test sample was a 5 mm wide sheet.

#### 2.4.4. Nuclear Magnetic Resonance Spectrometer Test

The AVANCE III 400 MHz solid-state nuclear magnetic resonance spectrometer produced by Bruker, Germany, was used to conduct ^29^Si NMR and ^27^Al NMR tests on the samples. The data were processed based on the Gauss–Lorentz method, and the relative intensity (I) of the characteristic peak was obtained by deconvolution with PeakFit software [14]. The average silicon chain length (MCL) of C-S-H gel in the sample and the degree of substitution of Al^3+^ for Si^4+^ (Al(4)/Si) on the silicon oxygen chain were quantitatively calculated by the following Formulas (4) and (5) [15].
(4)$$\mathrm{MCL}=\frac{2\left[I\left(Q^{1}\right){+I(Q}^{2}\left(0\mathrm{Al}\right))+1.5I\left(Q^{2}\left(1\mathrm{Al}\right)\right)\right]}{I\left(Q^{1}\right)}$$
(5)$$\mathrm{Al}\left[4\right]/\mathrm{Si}=\frac{I\left(Q^{2}\left(1\mathrm{Al}\right)\right)}{2\left[I\left(Q^{1}\right){+I(Q}^{2}\left(0\mathrm{Al}\right))+I\left(Q^{2}\left(1\mathrm{Al}\right)\right)\right]}$$
where, I(Q^1^) represents the relative intensity of the absorption peak corresponding to the [SiO_4_] tetrahedron at the end of the silica chain in the C-S-H gel structure of the hydrated paste; I(Q^2^(0Al)) represents the relative intensity of the absorption peak corresponding to the [SiO_4_] tetrahedron connected to two [SiO_4_] tetrahedrons on the straight chain of the C-S-H gel in the hydrated paste; I(Q^2^(1Al)) represents the relative intensity of the absorption peak corresponding to the [SiO_4_] tetrahedron adjacent to an [AlO_4_] tetrahedron on the straight chain of the C-S-H gel in the hydrated paste. The test sample is the sample powder passing 200-mesh sieve.

#### 2.4.5. Nitrogen Adsorption Test

The ASAP2020 full-automatic fast specific surface area and porosity analyzer provided by McMurray Teck Instrument Co., Ltd. (Fort McMurray, AB, Canada) was used to test the specific surface area and pore distribution of the sample. The specific surface area of BET was analyzed using Langmuir surface area model, with the lowest resolution of 0.01 m^2^/g. The Dubin–Radushkevich micropore area model was used for pore size distribution test, and the range was 0.35–500 nm. The test sample was a 5 mm cube.

## 3. Results and Discussion

### 3.1. Degree of Hydration

Figure 2 shows the ^29^Si NMR spectra of C_3_S-C_3_A paste after being hydrated for one day and then etched in 5wt% Na_2_SO_4_ solution for 7 d, 28 d, and 180 d. The deconvolution calculation results are shown in Table 3. Figure 3 shows the change curve of hydration degree of C_3_S-C_3_A paste under sulfate attack, and that the hydration degree of pure C_3_S paste increased by 9.98%, 5.02%, and 2.84%, respectively, after 7, 28, and 180 d. With the increase of C_3_A content, the hydration degree of composite paste increased. When the content of C_3_A was 20%, the hydration degree of the composite paste increased by 10.25%, 4.03%, and 4.17% at 7 d, 28 d, and 180 d, respectively. This shows that sulfate attack can promote the hydration process of C_3_S-C_3_A paste, thus improving the hydration degree of composite paste. In addition, sulfate attack promotes the hydration of C_3_S-C_3_A paste mainly in the early stage of hydration.

### 3.2. MCL

The MCL of the sample was calculated from Table 3 and Formula 1, and the results are shown in Figure 4. It can be seen from Figure 4 that compared with the noneroded sample, the MCL of C-(A)-S-H gel of pure C_3_S paste increased by 6.95%, 7.61%, and 10.95%; and the MCL of C_3_S-C_3_A paste with 20% C_3_A increased by 9.36%, 4.77%, and 6.39%, respectively, after 7, 28, and 180 d of Na_2_SO_4_ erosion. This indicates that sulfate attack could improve the polymerization degree of C_3_S-C_3_A paste, especially if the average silicon chain length in the early hydration stage is greater than that in the late hydration stage (180 d). From the analysis of the influence of sulfate attack on the hydration degree of C_3_S-C_3_A paste, it can be seen that sulfate attack can promote the hydration of composite paste, making the [SiO4] tetrahedron in the paste change from monomer to dimer and polymer structure, thus improving the MCL of C-(A)-S-H gel of composite paste. In addition, SO_4_^2−^ introduced by sulfate attack reacts with Ca(OH)_2_ in the composite paste to form gypsum and other products, which reduces the concentration of Ca^2+^ in the pore solution of paste, leading to the removal of Ca^2+^ between the layers of C-(A)-S-H gel, which is represented by the reduction of Ca/Si in C-(A)-S-H gel. The research shows that the decrease of Ca/Si will promote the transformation of [SiO4] tetrahedron in cement paste C-(A)-S-H gel from dimer to polymer and increase the length of the silica chain [16]. Therefore, sulfate attack improved the MCL of C_3_S-C_3_A paste C-(A)-S-H gel.

### 3.3. Al(4)/Si

Figure 5 demonstrates that compared with the noneroded sample, the Al(4)/Si of C-(A)-S-H of C_3_S-C_3_A paste with 5%, 10%, and 20% C_3_A content decreased by 17.39%, 16.22%, and 14.89% respectively after 7 d of Na_2_SO_4_ erosion; after 28 d of Na_2_SO_4_ erosion, Al(4)/Si of C-S-H decreased by 17.64%, 16.67%, and 17.95, respectively; and after 180 d of Na_2_SO_4_ erosion, the Al(4)/Si of C-S-H decreased by 33.33%, 38.01%, and 29.03%, respectively. This indicates that Na_2_SO_4_ erosion can lead to the decrease of Al(4)/Si of C-(A)-S-H gel of C_3_S-C_3_A paste, that is, dealumination. This is because when Al replaces Si on the silicon oxygen chain of C-(A)-S-H gel, the stability of the system will be reduced. At this time, the Al-O bond on the generated C-(A)-S-H gel is longer than the Si-O bond before substitution, making the Al atom in the metastable state, which is easier to detach under the erosion of Na_2_SO_4_, forming other aluminum phase products [17]. Therefore, with the extension of the erosion age, Al(4)/Si decreases. 

In order to further explore the effect mechanism of C_3_A content on Al(4)/Si of C_3_S-C_3_A paste (see Formula 6), the sample was tested by ^27^Al NMR. Figure 6 shows ^27^Al NMR spectra of C_3_S-C_3_A paste with 5%, 10%, and 20% C_3_A content after being hydrated for one day and then eroded in 5wt% Na_2_SO_4_ solution for 7 d, 28 d, and 180 d. Table 4 shows the deconvolution calculation results of ^27^Al NMR spectra.
TAH + SO_4_^2−^ ⇌ AFt + AFm(6)

Taking C_3_S-C_3_A paste with a content of 20% C_3_A as an example, the Al(4) content of the paste decreased by 2.49% and 5.75% after being eroded by Na_2_SO_4_ for 7 d and 180 d, respectively, compared with the noneroded sample and the AFt content increased by 3.27% and 4.96%, respectively; the relative content of AFm decreased by 4.67% and 25.09% respectively; and the TAH decreased by 5.54% and 7.80%, respectively. The analysis results show that: (1) Sulfate attack caused the relative content of Al(4) in C_3_S-C_3_A paste to decrease, which was consistent with the analysis results of the ^29^Si NMR spectrum; (2) sulfate attack led to an increase in the relative content of AFt and a decrease in the relative content of AFm and this was because there was enough SO_4_^2−^ in the erosion solution to support the conversion of Al(4) from C-(A)-S-H gel to AFt instead of AFm; and (3) sulfate attack could also reduce the relative content of TAH, because TAH and SO_4_^2−^ can transform with AFm and AFt at room temperature [18]. Under sulfate attack, the chemical balance moved to the right, resulting in a decrease in the relative content of TAH.

### 3.4. Ca/Si

The C_3_S-C_3_A paste eroded in 5 wt% Na_2_SO_4_ solution for 28 d after hydration for 1 d was observed by a scanning electron microscope and energy dispersive X-ray spectrometer, and 20 points were selected on the C-(A)-S-H gel of each sample for element analysis. The average Ca/Si of composite paste C-(A)-S-H gel was calculated according to the results, as shown in Figure 7.

According to SEM–EDS, Ca/Si of C_3_S-C_3_A paste C-(A)-S-H gel after sulfate attack still increases with the increase of C_3_A content. In addition, compared with the SEM–EDS map of the noneroded C_3_S-C_3_A paste, the Ca/Si of C_3_S-C_3_A paste with 0%, 10%, and 20% C_3_A content decreased by 12.81%, 14.29%, and 16.16% respectively after sulfate attack. This shows that sulfate attack will lead to the decrease of Ca/Si of C_3_S-C_3_A paste, and the decrease range will increase with the increase of initial Ca/Si of the paste. This is because SO_4_^2−^ in Na_2_SO_4_ solution brings Ca^2+^ between layers or interfaces of C-(A)-S-H gel into solution through a diffusion adsorption desorption process to form gypsum and combines with Al phase hydration products in paste to further generate AFm or AFt, leading to the decalcification of C-(A)-S-H gel. The degree of decalcification of the composite paste C-(A)-S-H gel is related to its initial Ca/Si. Ding et al. [19] studied the preparation of C-(A)-S-H gel with Ca/Si from high to low using hydrothermal synthesis and studied the effect of sulfate attack on the synthesized C-(A)-S-H gel and Ca/Si of gelled paste. The results showed that the degree of decalcification of sulfate on C-(A)-S-H gel increased with the increase of initial Ca/Si of C-(A)-S-H gel. The addition of C_3_A led to the increase of interlayer Ca^2+^ of C_3_S-C_3_A paste C-(A)-S-H gel, which reduced the thermodynamic stability of the Ca-O layer of C-(A)-S-H gel. Therefore, the decalcification amplitude of C_3_S-C_3_A paste C-(A)-S-H gel under sulfate attack increased with the increase of C_3_A content.

### 3.5. Composition of Hydration Products

It can be seen from Figure 8 that the hydrated products of C_3_S-C_3_A paste under sulfate attack mainly include gypsum, AFt, Ca(OH)_2_, and some C_3_S and C_3_A that have not been completely hydrated. With the extension of erosion age, the intensity of C_3_S and C_3_A diffraction peaks in the XRD patterns of the paste decreased gradually, and basically disappeared at 180 d. Compared with XRD patterns of noneroded C_3_S-C_3_A composite hydrated paste, it was found that the diffraction peak of gypsum in the composite paste under sulfate attack increased with the increase of erosion age. Moreover, the second peak of AFt (close to 14°) disappeared on the 7th and 28th days. This was due to the reaction of C_3_A and sulfate ion to generate AFt at the initial stage of hydration. However, with the progress of hydration reaction, the invasion rate of sulfate ion was less than its consumption rate in chemical reaction. Therefore, the undersaturation of sulfate ions in the paste increased, and AFt was converted to AFm. With the erosion taking place, the defects in the paste increased, and the sulfate ion content increased accordingly. Therefore, the peak of AFt at 14° appeared again in the 180 d XRD spectrum. In addition, the apparent AFt diffraction peak appeared in the late attack period (28–180 d), and the intensity of AFt peak increased with the increase of erosion age. It showed that SO_4_^2−^ introduced by sulfate attack will react with the hydration products of C_3_S-C_3_A paste to generate gypsum and AFt, thus causing the change of C_3_S-C_3_A paste composition.

### 3.6. Pore Characterization

According to the BET nitrogen adsorption test results of C_3_S-C_3_A paste, the nitrogen adsorption desorption isotherm of C_3_S-C_3_A paste formed by single mineral hydration had a desorption hysteresis loop, which belonged to type IV adsorption desorption curve. It can be seen from Table 5 that with the increase of C_3_A content from 0% to 20%, the hysteresis loop became more obvious; the specific surface area of the slurry increased from 6.1999 m^2^/g to 8.6742 m^2^/g, with an increase of 39.91%; and the cumulative pore volume increased from 0.038636 mL/g to 0.044416 mL/g, with a growth rate of 4.83%. However, the average aperture decreased from 14.3549 nm to 11.5584 nm, with a decrease of 19.48%. Figure 9 and Figure 10 show that the addition of C_3_A will introduce mesopores with pore diameter of ≤ 20 nm into the composite paste. With the increase of C_3_A content, the specific surface area and cumulative pore volume of the composite paste increased, and the average pore diameter decreased. The final performance is the increase of the total porosity of the composite paste.

Figure 11 and Figure 12 show the nitrogen adsorption test results of C_3_S-C_3_A paste under sulfate attack. Table 6 demonstrates that the specific surface area of C_3_S-C_3_A paste with 10% C_3_A content after sulfate attack decreased by 7.05%; the cumulative pore volume decreased by 20.39%; and the average pore diameter decreased by 14.95%. This indicates that sulfate attack generates gypsum, Aft, and other products in the gel pores of the paste, resulting in the reduction of the specific surface area, cumulative pore volume, and average pore diameter of the paste.

## 4. Conclusions

In this paper, the effect of the C_3_A content on the structural characterization of C_3_S-C_3_A paste under sulfate attack was found through indoor immersion. The structural characterization of C_3_S-C_3_A paste was directly related to its C_3_A content. From the experimental results, the following conclusions can be drawn:Sulfate attack can improve the MCL of C-(A)-S-H gel in the C_3_S-C_3_A paste by promoting the hydration of the paste. With the increase of C3A content, the effect of sulfate ion on hydration is more significant.Sulfate erosion prevents Al^3+^ formed by C_3_A hydration from entering the C-(A)-S-H gel bridge site, resulting in the decrease of Al(4)/Si in C-(A)-S-H gel. The decrease amplitude of Al(4)/Si increased with the increase of C_3_A content.With the prolongation of sulfate attack age, Ca/Si of C-(A)-S-H gel decreased. Sulfate attack had a decalcification effect on C-(A)-S-H gel formed by C_3_S-C_3_A hydration, leading to its microstructure evolution. The decrease amplitude of Ca/Si also increased with the increase of C_3_A content.The SO_4_^2−^ intruded into the paste reacts with the Al(4) migrated from the silica chain bridge of C-(A)-S-H gel and the Al containing phase hydration products (AFm and TAH) in the paste to generate AFt, resulting in the increase of the relative content of AFt in C_3_S-C_3_A paste.The addition of C_3_A reduced the average pore diameter of C_3_S-C_3_A paste and increased the porosity of the paste. Sulfate attack reduced the pore specific surface area, cumulative pore volume, and average pore diameter of C_3_S-C_3_A paste.The minerals contained in concrete structures in practical engineering applications are not limited to C_3_S and C_3_A. In the future, we will consider conducting experiments on mineral phases such as C_4_AF and C_2_S. Through the test data, the mineral composition of the cementitious system was designed to prepare concrete structures that can achieve practical engineering application.

## Figures and Tables

**Figure 1 materials-16-00077-f001:**
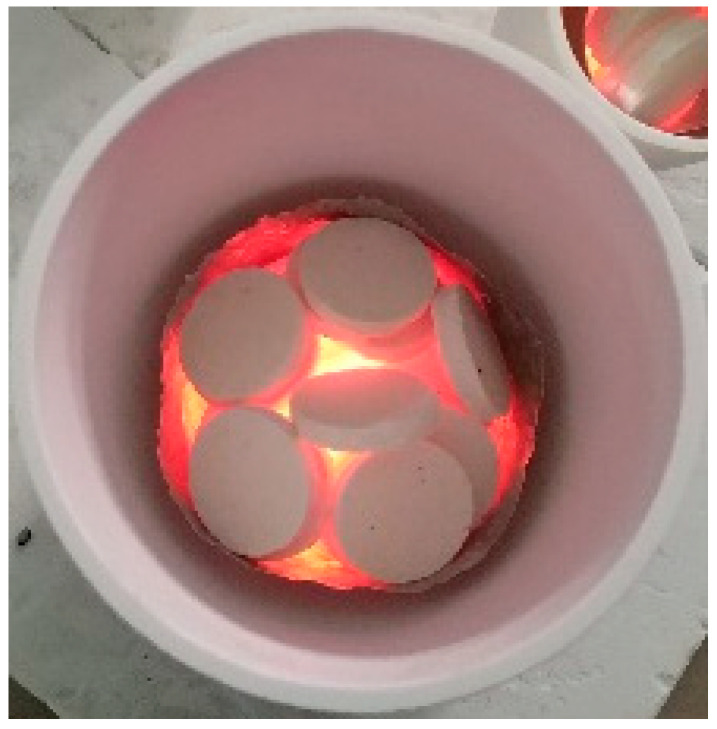
Calcined sample.

**Figure 2 materials-16-00077-f002:**
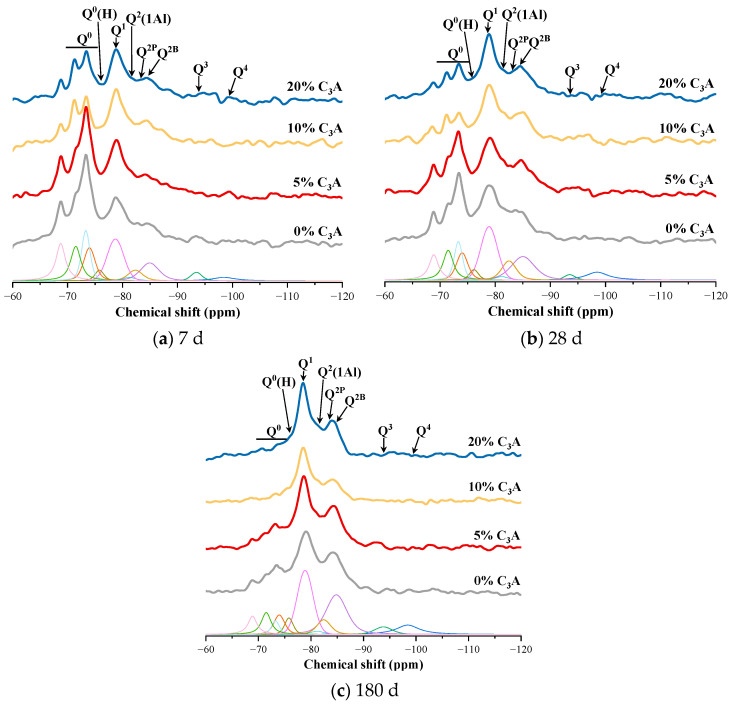
^29^Si NMR spectra of C_3_S-C_3_A paste under sulfate attack.

**Figure 3 materials-16-00077-f003:**
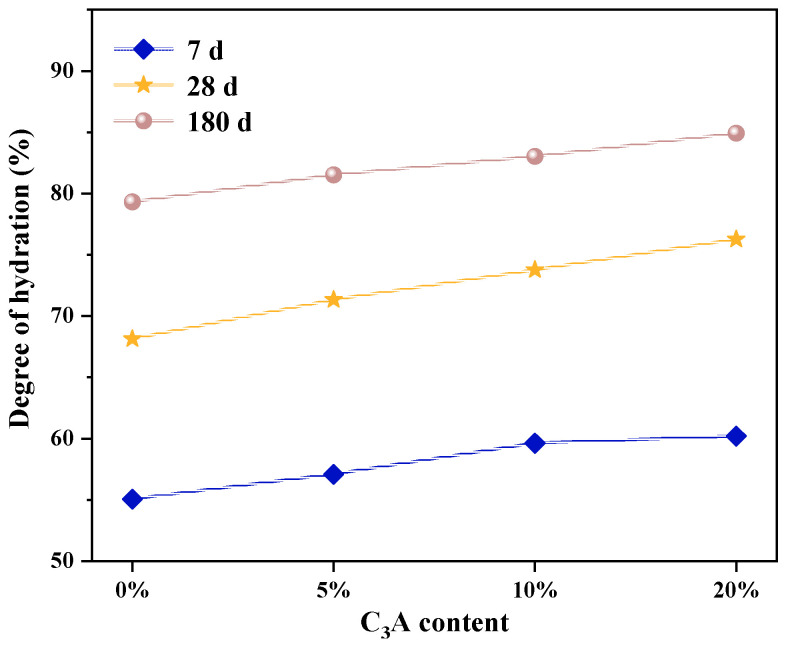
Degree of hydration of C_3_S-C_3_A paste under sulfate attack.

**Figure 4 materials-16-00077-f004:**
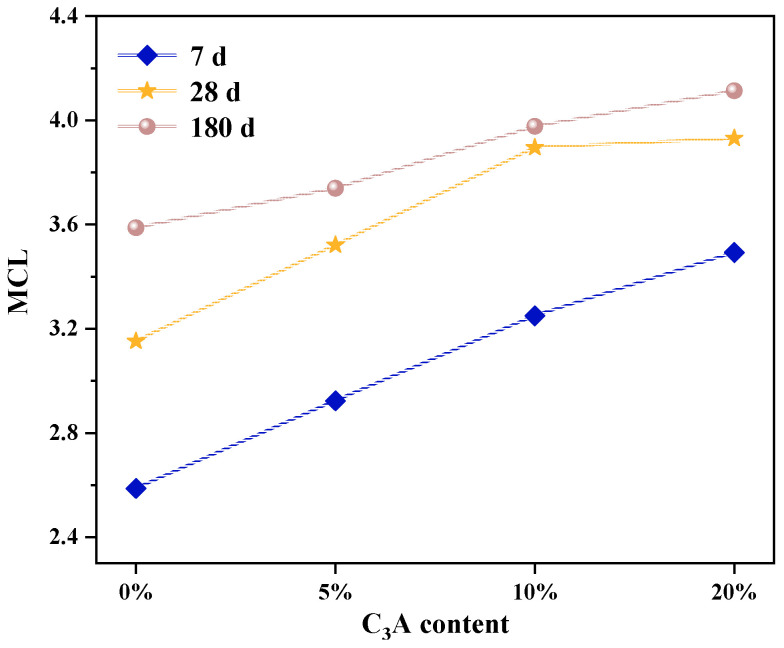
MCL variation of C-S-H gel in C_3_S-C_3_A paste under sulfate attack.

**Figure 5 materials-16-00077-f005:**
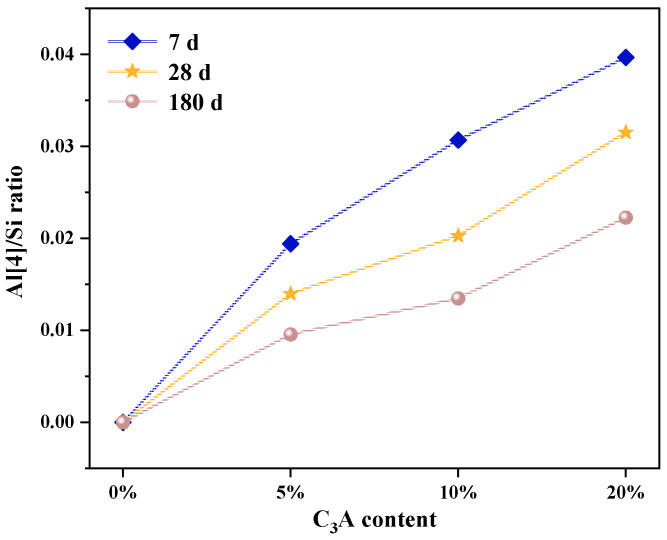
Al(4)/Si variation of C-S-H gel in C_3_S-C_3_A paste under sulfate attack.

**Figure 6 materials-16-00077-f006:**
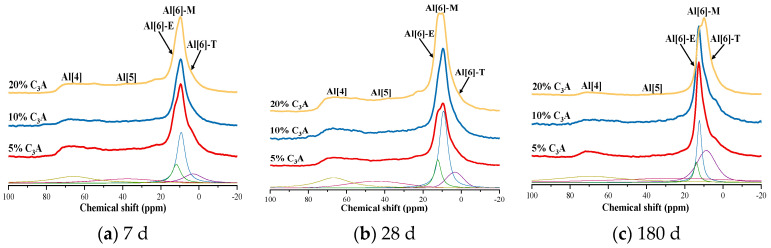
^27^Al NMR spectra of C_3_S-C_3_A paste under sulfate attack.

**Figure 7 materials-16-00077-f007:**
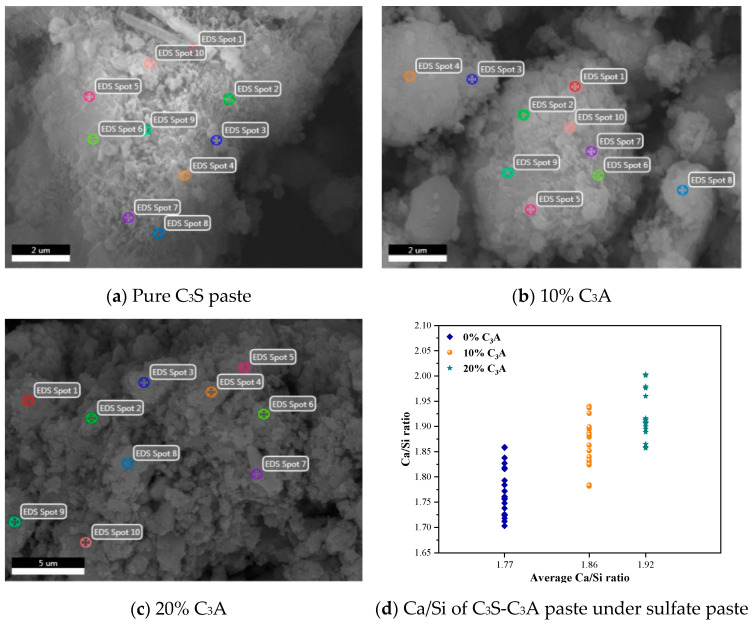
SEM–EDS spectra of C_3_S-C_3_A paste under sulfate attack.

**Figure 8 materials-16-00077-f008:**
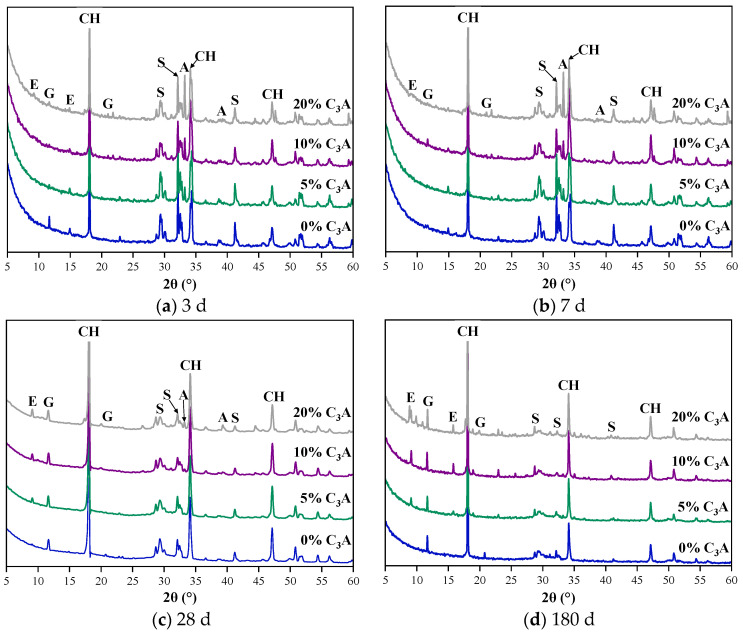
XRD NMR spectra of C_3_S-C_3_A paste under sulfate attack, where G = CaSO_4_·2H_2_O, E = AFt, CH = Ca(OH)_2_, and S = C_3_S, A = C_3_A.

**Figure 9 materials-16-00077-f009:**
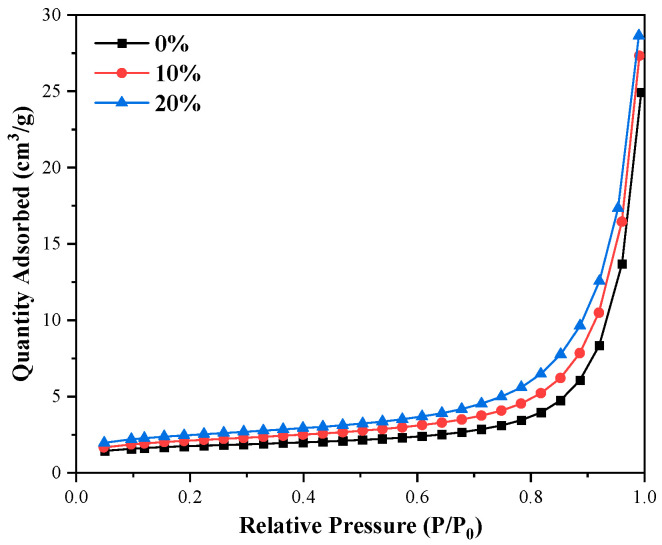
N_2_ adsorption desorption isotherm of C_3_S-C_3_A paste.

**Figure 10 materials-16-00077-f010:**
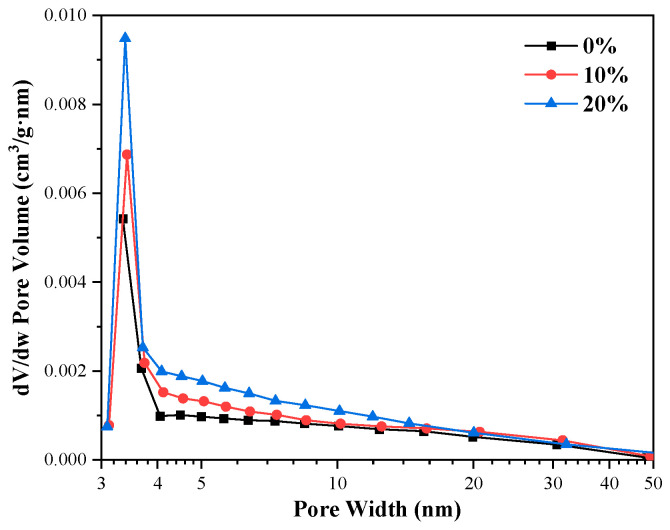
Pore diameter distribution of C_3_S-C_3_A paste.

**Figure 11 materials-16-00077-f011:**
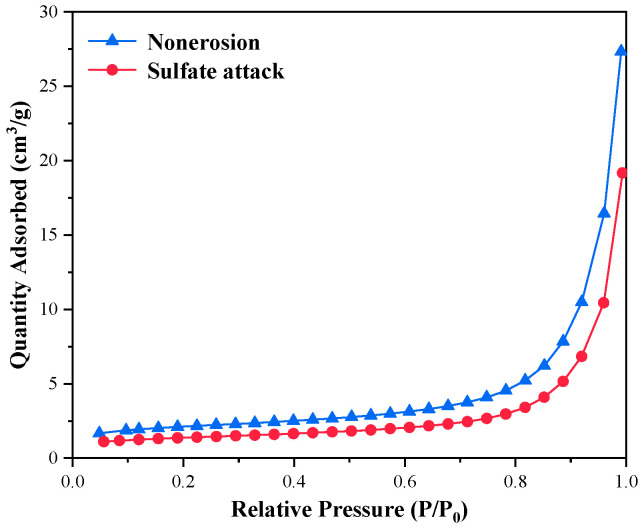
N_2_ adsorption desorption isotherm of C_3_S-C_3_A paste under sulfate attack.

**Figure 12 materials-16-00077-f012:**
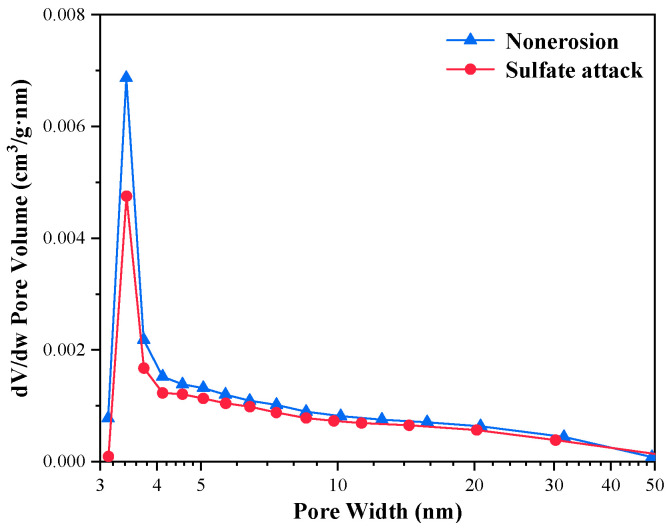
Pore diameter distribution of C_3_S-C_3_A paste under sulfate attack.

**Table 1 materials-16-00077-t001:** Mixture proportion of C_3_A-C_3_S paste.

No.	w/b	C_3_A	C_3_S	CaSO_4_·2H_2_O
		(kg/m^3^)	(kg/m^3^)	(kg/m^3^)
1	0.5	0	1000	50
2	0.5	50	950	50
3	0.5	100	900	50
4	0.5	200	800	50

**Table 2 materials-16-00077-t002:** f-CaO content in single mineral. (%).

Single Mineral Type	After the First Calcination	After the Second Calcination	After the Third Calcination
C_3_A	6.41	1.08	0.56
C_3_S	2.35	1.17	0.73

**Table 3 materials-16-00077-t003:** Deconvolution Results of ^29^Si NMR Spectra.

C_3_A Content (%)	Age (d)	Q^n^ Relative Strength Value I (%)						*a_C_* (%)
		Q^0^	Q^0H^	Q^1^	Q^2^(1Al)	Q^2B^	Q^2P^	
0.0	7	44.927	1.484	41.430	0.000	4.132	8.028	55.07
5.0		42.907	5.900	35.706	1.988	4.522	8.977	57.09
10.0		40.379	6.098	33.953	3.284	5.461	10.826	59.62
20.0		39.795	5.494	32.573	4.342	5.913	11.883	60.21
0.0	28	31.866	9.321	37.319	0.000	7.111	14.382	68.13
5.0		28.668	7.106	36.987	1.794	8.462	16.983	71.33
10.0		26.240	7.228	34.851	2.699	9.700	19.281	73.76
20.0		23.738	8.281	35.676	4.285	9.345	18.676	76.26
0.0	180	20.690	9.358	38.998	0.000	10.477	20.477	79.31
5.0		18.486	6.581	40.460	1.430	10.906	22.137	81.51
10.0		16.981	6.241	39.129	2.067	11.899	23.684	83.02
20.0		15.084	3.644	40.395	3.616	12.424	24.836	84.92

**Table 4 materials-16-00077-t004:** Deconvolution Results of ^27^Al NMR Spectra.

C_3_A Content (%)	Age (d)	Al^3+^ Relative Strength Value I(%)				
		Al(4)	Al(5)	Al(6)-E	Al(6)-M	Al(6)-T
5.0	7	27.198	11.586	26.222	29.028	5.966
10.0		17.897	13.849	30.905	26.493	9.857
20.0		17.527	16.885	33.579	27.856	4.153
5.0	28	21.662	10.263	28.787	30.066	8.241
10.0		16.789	11.130	31.648	31.136	9.296
20.0		13.823	11.764	36.065	28.974	9.373
5.0	180	10.947	4.454	47.052	26.338	11.210
10.0		8.571	3.967	51.009	23.833	12.620
20.0		3.740	2.793	57.078	22.823	13.566

**Table 5 materials-16-00077-t005:** Pore characterization of C_3_S-C_3_A paste.

C_3_A Content (%)	Specific Surface Area (m^2^·g^−1^)	Cumulative Pore Volume (mL·g^−1^)	Average Pore Diameter (nm)
0	6.1999	0.038636	14.3549
10	7.4032	0.042371	13.2907
20	8.6742	0.044416	11.5584

**Table 6 materials-16-00077-t006:** Pore characterization of C_3_S-C_3_A paste under sulfate attack.

	Specific Surface Area (m^2^·g^−1^)	Cumulative Pore Volume (mL·g^−1^)	Average Pore Diameter (nm)
Noneroded	7.4032	0.042371	13.2907
Sulfate attack	6.8815	0.033731	11.3044

## Data Availability

Data are contained within the article.

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
