# Peer review of "Influence of Structural Characterization of C3S-C3A Paste under Sulfate Attack"

_materials, 2022, doi:10.3390/ma16010077_

Round 1

Reviewer 1 Report

This paper aimed to examine the effect of sulfate attack on the formation and evolution of hydration products and microstructures of C3A-C3S pastes manufactured with different C3A content. Please revise the manuscript by reflecting my commnets below. 

0. All Part

Need to check chapter number. 

1. abstract

- It is necessary to mention the effect of sulfate attack on the formation and evolution of hydration products and microstructures of C3A-C3S paste prepared with different C3A content.

2.1. Materials

- More information about the material is required. (e.g. particle size and purity, etc.)

2.3. Preparation and curing of C3A-C3S paste

- A more detailed explanation of the hydration stop process is needed.

- In addition, ethyl ether is generally used in the hydration stop process to remove the organic solvent remaining in the sample. However, why was not used ethyl ether?

2.4.2. X-ray diffraction test

- Additional explanation of which program was used for XRD analysis is required.

3.5. Composition of hydration products

- As seen in the XRD patterns of the 20% C3A sample, AFt peak (near 14 degrees) disappeared on the 7 and 28 days but it appeaed again on the 180 days. Therefore, it is necessary to add discussions related to this part.

3.6. Pore characterization

-The purpose of this paper is to find out the structural characteristics of C3S-C3A paste. 

Therefore, it is necessary to provide pore structure data for all C3S-C3A pastes with 0%, 10%, and 20% C3A content and then perform a comparative analysis.

4. Conclusions

Conclusions need to be rewritten based on additionally analyzed experimental results.

Author Response

Comment 1: Need to check chapter number.

Answer: Thank you very much for your comments. We have revised the chapter numbers in the manuscript and checked the full text to avoid other format problems.

Comment 2: It is necessary to mention the effect of sulfate attack on the formation and evolution of hydration products and microstructures of C3A-C3S paste prepared with different C3A content.

Answer: Thank you very much for your comments. We have added in the abstract the influence of sulfate attack on the formation and evolution of hydration products and microstructure of C3A-C3S slurry prepared with different C3A content.

Comment 3: More information about the material is required. (e.g. particle size and purity, etc.)

Answer: Thank you very much for your comments. We have included information about raw materials in the manuscript.

Comment 4: A more detailed explanation of the hydration stop process is needed.

Answer: Thank you very much for your comments. We have described the method of sample hydration termination in more detail in the manuscript.

Comment 5: In addition, ethyl ether is generally used in the hydration stop process to remove the organic solvent remaining in the sample. However, why was not used ethyl ether?

Answer: Thank you very much for your comments. Anhydrous ethanol, ethyl ether and isopropanol are widely used as organic solvents to stop the hydration of cement paste. Due to ethyl ether is easy to be oxidized, and considering the problem of test accuracy, absolute ethanol was finally selected in this experiment.

Comment 6: Additional explanation of which program was used for XRD analysis is required.

Answer: Thank you very much for your comments. We have included a description of the procedure used for XRD analysis in the manuscript.

Comment 7: As seen in the XRD patterns of the 20% C3A sample, AFt peak (near 14 degrees) disappeared on the 7 and 28 days but it appeaed again on the 180 days. Therefore, it is necessary to add discussions related to this part.

Answer: Thank you very much for your comments. We have added a discussion on this part to the manuscript.

Comment 8: The purpose of this paper is to find out the structural characteristics of C3S-C3A paste. Therefore, it is necessary to provide pore structure data for all C3S-C3A pastes with 0%, 10%, and 20% C3A content and then perform a comparative analysis.

Answer: Thank you very much for your comments. We have provided the pore structure data of all C3S-C3A paste with 0%, 10% and 20% C3A content in the manuscript, and carried out comparative analysis.

Comment 9: Conclusions need to be rewritten based on additionally analyzed experimental results.

Answer: Thank you very much for your comments. We have modified the conclusion according to the test results after re analysis.

Reviewer 2 Report

Article entitled as "Influence of structural characterization of C3S - C3A paste under sulfate attack" discuss about the paste modification under sulfate attack. I accept to publish the article under minor revisions. Authors are requested to update the article as per following comments

1. Authors are requested to remove nomenclature in the title and in keywords

2. In abstract, try to introduce the full name before nomenclature. For example, at first introducing Calcium Silicate Hydrate, the full name should be first introduced, followed by nomenclature as CSH

3. Authors are requested to check the nomenclature of the article throughout the text

4. Authors are requested to add more information in the introduction. 11 references with two paragraphs are enough for a quality magazine? Try to add two more paragraphs in introduction

5. Research gap is not clear. Kindly rewrite it

6. Research objectives are confusing. Make it as in simple sentences

7. Section 2.1 is not clear. Introduce the physical and chemical properties of materials used for investigation

8. 2.2 cement single ore? Try to explain it under its title. Try to explain the method used with proper references. Rewrite section 2.2

9.include references for section 2.3

10. In table 1, try to represent ingredients in same unit (for example kg/m3. Include mix id name in it.

11. Remove the name formula. Just use equation number alone

12. Check the heading 2.3.4. use just nuclear magnetic resonance spectrometer.

13. Include references for section 3.1

14. Include sample preparation for the experiments like XRD, SEM, etc

15. Don't use nomenclature for the table and figure titles

16. MCL? Nomenclature? 

17. More references are needed for results and discussion part

18. Try to keep figure in single page (for example figure 7)

19. Several sentences in results and discussion part are too long. Try to make it as simple. For example page 9, first paragraph

20. Future scope of the study and application part should be included in conclusion part

Author Response

Comment 1: Authors are requested to remove nomenclature in the title and in keywords. 

Answer: Thank you very much for your comments. We have changed the title and deleted some keywords as required.

Comment 2: In abstract, try to introduce the full name before nomenclature. For example, at first introducing Calcium Silicate Hydrate, the full name should be first introduced, followed by nomenclature as CSH.

Answer: Thank you very much for your comments. We have introduced the full name of the CSH. In addition, we checked the full text to avoid similar problems.

Comment 3: Authors are requested to check the nomenclature of the article throughout the text.

Answer: Thank you very much for your comments. We have revised the chapter numbers in the manuscript and checked the full text to avoid other format problems.

Comment 4: Authors are requested to add more information in the introduction. 11 references with two paragraphs are enough for a quality magazine? Try to add two more paragraphs in introduction. 

Answer: Thank you very much for your comments. We have added new content to the introduction.

Comment 5: Research gap is not clear. Kindly rewrite it.

Answer: Thank you very much for your comments. We have re described the research gap.

Comment 6: Research objectives are confusing. Make it as in simple sentences.

Answer: Thank you very much for your comments. We have restated our research objectives.

Comment 7: Section 2.1 is not clear. Introduce the physical and chemical properties of materials used for investigation. 

Answer: Thank you very much for your comments.

Comment 8: 2.2 cement single ore? Try to explain it under its title. Try to explain the method used with proper references. Rewrite section 2.2. 

Answer: Thank you very much for your comments. “cement single ore” should be “single mineral of cement clinker”. We have revised the inappropriate statement about “cement single ore” in the full text.

Comment 9: include references for section 2.3. 

Answer: Thank you very much for your comments. We have revised the inappropriate statement about “cement single ore” in the full text.

Comment 10: In table 1, try to represent ingredients in same unit (for example kg/m3. Include mix id name in it. 

Answer: Thank you very much for your comments. We have unified the units in Table 1.

Comment 11: Remove the name formula. Just use equation number alone. 

Answer: Thank you very much for your comments. We have deleted the name formula.

Comment 12: Check the heading 2.3.4. use just nuclear magnetic resonance spectrometer. 

Answer: Thank you very much for your comments. We have revised the title as required.

Comment 13: Include references for section 3.1. 

Answer: Thank you very much for your comments. We have revised the inappropriate statement about “cement single ore” in the full text.

Comment 14: Include sample preparation for the experiments like XRD, SEM, etc. 

Answer: Thank you very much for your comments. We have added the preparation instructions for XRD, SEM and other test samples.

Comment 15: Don't use nomenclature for the table and figure titles. 

Answer: Thank you very much for your comments. We have deleted the term as much as possible, but C3A, as a variable, has to appear in some charts.

Comment 16: MCL? Nomenclature? 

Answer: Thank you very much for your comments. MCL has been described in Section 2.4.4 when it first appeared.

Comment 17: More references are needed for results and discussion part. 

Answer: Thank you very much for your comments. We have modified the conclusion according to the test results after re analysis.

Comment 18: Try to keep figure in single page (for example figure 7). 

Answer: Thank you very much for your comments. We have adjusted figures to a single page.

Comment 19: Several sentences in results and discussion part are too long. Try to make it as simple. For example page 9, first paragraph. 

Answer: Thank you very much for your comments. We have simplified the sentence pattern of the conclusion.

Comment 20: Future scope of the study and application part should be included in conclusion part. 

Answer: Thank you very much for your comments. We have included a vision of the future at the end of the manuscript.

Reviewer 3 Report

The work is of an experimental nature related to the durability of cement materials subjected to the influence of sulfate ions.

The authors tested the durability of the C3S-C3A paste with different C3A content exposed to sulfate ions on the 3rd, 7th, 28th and 180th day.

The composition of corrosion products (Ca/Si, Al/Si and C-(A)-S-H gel) as well as the formation of microstructure, pore structure, migration and phase changes were analyzed. It has been shown that the interaction of sulfate ions can induce hydration in the C3S-C3A paste and accelerate the formation of C-(A)-S-H gel as well as its dealimination and decalcification. At the same time, expanded products were created - ettringite and gypsum. The structure tests showed a decrease in the specific surface area, cumulative surface area and mean pore diameter.

I evaluate my work very positively. The performed research is valuable from the cognitive point of view. I negatively assess the lack of explanation of some actions performed by the authors.

Here are my detailed comments:

1. Chapter 1: It is appropriate to explain where and when sulfate ion attack occurs. Does this affect the cover properties of the reinforcement? Please highlight the purpose of the work and the novelties contained in the work.

2. Chapter 2.1: I suggest adding a comment on why these and not other materials were used.

3. Chapter 2.4: It would be clearer to include the poor test program in the table. Then the comments in the text are enough.

I suggest supplementing the information whether the samples were standardized, if not, why such and not another shape was used.

4. Chapter 3.1 - 3.5: The research results are very well and interestingly described. Basically, I have no comments.

5. Section 3.6: There is no explanation for changes in pore structure. I propose to extend the comment.

6. Chapter 4: Suggests adding directions for further work.

Author Response

Comment 1: Chapter 1: It is appropriate to explain where and when sulfate ion attack occurs. Does this affect the cover properties of the reinforcement? Please highlight the purpose of the work and the novelties contained in the work.

Answer: Thank you very much for your comments. We have emphasized the harm of sulfate ion in the introduction. In addition, we also emphasized the purpose of the work and the novelty contained in the work.

Comment 2: Chapter 2.1: I suggest adding a comment on why these and not other materials were used.

Answer: Thank you very much for your comments. We have explained the choice of materials.

Comment 3: Chapter 2.4: It would be clearer to include the poor test program in the table. Then the comments in the text are enough. I suggest supplementing the information whether the samples were standardized, if not, why such and not another shape was used.

Answer: Thank you very much for your comments. The mold required for sample preparation was selected according to reference 12.

Du Lang, Li Yuxiang, Ma Xue. Preparation and characterization of single mineral of cement clinker. Cement, 2014, (8): 15-18.

Comment 4: Chapter 3.1 - 3.5: The research results are very well and interestingly described. Basically, I have no comments.

Answer: Thank you very much for your comments.

Comment 5: Section 3.6: There is no explanation for changes in pore structure. I propose to extend the comment.

Answer: Thank you very much for your comments. We have provided the pore structure data of all C3S-C3A paste with 0%, 10% and 20% C3A content in the manuscript, and carried out comparative analysis.

Comment 6: Chapter 4: Suggests adding directions for further work.

Answer: Thank you very much for your comments. We have included a vision of the future at the end of the manuscript.
